# The Impact of Interactive Control in Budget Management on Innovation Performance of Enterprises: From the Perspective of Manager Role Stress

**DOI:** 10.3390/ijerph20032190

**Published:** 2023-01-25

**Authors:** Xiangfei Zeng, Ning Zhang, Lianghua Chen, Wenpei Zhang

**Affiliations:** 1School of Business, Anhui University of Technology, Ma’anshan 243032, China; 2School of Economics and Management, Southeast University, Nanjing 211100, China

**Keywords:** interactive control, individual psychology, role stress of manager, enterprise innovation performance, environmental turbulence

## Abstract

This study aimed to study the influence of the interactive budget on enterprise innovation performance from the perspective of managerial psychology, and to examine the mediating effect of managers’ stress and the moderating effect of environmental turbulence in this influence. The study collected 228 managers’ data in the Yangtze River Delta region of China through online questionnaires; the structural equation model is constructed by IBM SPSS Amos 24.0, and the data is analyzed by Mplus 7.4 and IBM SPSS 23.0. The results showed that: (1) an interactive budget positively affects enterprise innovation performance; (2) the role stress of managers plays a partial mediating role in the relationship between an interactive budget and enterprise innovation performance; (3) environmental turbulence positively moderates the relationship between the interactive budget and enterprise innovation performance; and (4) the positive effect of the interactive budget on enterprise innovation performance is more significant in enterprises with a high proportion of female managers, and the negative effect on the role stress of managers is more significant in high-tech enterprises. In theory, the results enrich the research on the impact of an interactive budget on individual managers’ psychology, and further reveals the “black box” of the impact of an interactive budget on enterprise innovation performance; in practice, the results make enterprises attach importance to the influence of control environments on the psychological health of managers, and provides a reference for enterprises to use control methods rationally to promote innovation in budget management.

## 1. Introduction

Innovation can enhance the flexibility and adaptability of enterprises and is a key element for enterprises to acquire and maintain core competitiveness [1]. In the process of enterprise innovation, management control plays an indispensable role [2]. In recent years, the relationship between management control systems and innovation has attracted much attention [2,3,4]. Management control systems play a key role in the allocation of resources in enterprises [4], which helps improve the level of enterprise innovation. The extensive application of management control systems can reduce potential risks, make the new product and development process more transparent and efficient, and thus enhance the enterprise’s innovation capability [5]. In today’s changing environment, how to better apply a management control system to promote innovation ability is of great significance for the improvement of labor productivity and core competitiveness of enterprises [6]. China’s economy is in a critical period of transformation and upgrading, which requires continuous strengthening of enterprise innovation to achieve high-quality development of the national economy. Exploring the intrinsic linkage between budget management and innovation in the Chinese context is important for the longevity of enterprises, and national survival and development in China, as well as in countries and regions around the world that are, or will be, going through a similar transformation.

Among many management control systems, budget management has received much attention because of its important role in organizational planning, resource allocation, and performance evaluation [7]. However, the traditional budget may limit managers’ acquisition of work resources [8], exacerbate managers’ conflicts between achieving budget goals and organizational innovation, and thus increase managers’ work stress [9], and is not conducive to organizational innovation decisions [10]. With the changing business environment, the turbulence of the external environment poses new challenges to budget management. When the level of environmental turbulence is high, the individuals’ innovation consciousness will increase [11], and the control system will also promote product innovation by increasing the information supply to adapt to environmental changes [2]. Correspondingly, senior managers will use interactive control in budget activities to meet the requirements of strategic uncertainty [12]. The interactive budget refers to a budget method that uses interactive control in the budget process [13]. Interactive control is a way to use the control system, when senior managers use the control system interactively, meaning that they will regularly participate in subordinate decisions, discuss with subordinates in face-to-face communication mode, and pay more attention to the risk information related to enterprise strategy [13]. Decision-making under traditional budgets is relatively centralized [8], leading enterprises to focus too much on the cost and results of the budget [8]. Moreover, it is difficult with the traditional budget to adjust the budget target in response to the external environment, making the budget prepared by the enterprise a “castle in the air”, resulting in a mismatch between the budget and the enterprise strategy [14]. However, through the interactive budget, managers focus their attention on budget risk information by obtaining a large amount of effective business information on a fully interactive basis [13] so that they can have sufficient resources to deal with the innovative activities of enterprises [15]. Therefore, interactive budgets can create a well-resourced working environment for managers, stimulating them to generate new ideas and inspiring their motivation and initiative [15]. Relevant scholars have studied the effects of interactive budgets on enhancing information flow, promoting knowledge absorption, coping with strategic uncertainty, and shaping psychological capital; interactive budgets allows managers to generate dialogue and encourage information sharing in the budgeting process [16], and can improve the flow of information and create positive information environment [17]; interactive budgets can facilitate the exchange of hidden knowledge, promoting organizations to identify opportunities and improve performance [18]; interactive budgets can effectively stimulate the double-loop learning in the team, thus solving the strategic uncertainty [19]; interactive budgets promote team efficiency by enhancing individual perceived team effectiveness [20], and it can also address the role ambiguity of up and down to cope with environmental uncertainties [21].

However, there are still issues that can be explored in the study of the relationship between interactive budget control and enterprise innovation: (1) interactive budgets have not been deeply studied at the level of individual psychology. In recent years, in the field of management research, more and more attention has been paid to the research involving the organizational level and the individual level [22], but the current research on interactive budgets mainly stays at the level of companies and organizations, with little research on the level of individual psychology; also, studies on interactive budgets mainly focused on the demographic characteristics of managers [23], rarely involving individual psychological aspects, such as stress, and ignoring the managers’ own perceived initiative. (2) Interactive budgets can promote enterprise innovation [16,24]; however, there is a lack of research on the mechanism between them, which is not conducive to our comprehensive understanding of the innovation effect of an interactive budget. In addition, the studies on the impact of budget management on enterprise innovation in the Chinese context is also quite scarce.

This study introduces the idea of the psychology theory in contingency theory, developed from the traditional contingency theory, which developed the hypothesis of the environment as an objective external factor affecting the organization to the hypothesis of the environment as an interactive factor of individual perception. Specifically, based on the demand-resource (JD-R) model, this study empirically analyzed the impact of interactive budgets on enterprise innovation performance in China’s Yangtze River Delta region, and explored the mediating role of manager role stress and moderating role of environmental turbulence in the relationship between the interactive budget and enterprise innovation performance. The following four issues will be investigated: (1) can interactive budgets improve enterprise innovation performance? (2) does the role stress level of managers mediate the impact of interactive budgets on enterprise innovation performance? (3) will environmental turbulence affect the relationship between interactive budgets and enterprise innovation performance? (4) does the innovation effect of interactive budgets and its effect on the managers’ role stress differ due to enterprise heterogeneity (i.e., managerial gender, industry type, etc.)? The possible contribution of this study may include: theoretically, it enriches the research on the psychological impact of interactive control on individual managers, and further opens the “black box” of the impact of interactive control in budget management on enterprise innovation performance; in practice, it is helpful for enterprise leaders to pay attention to managers’ psychological conditions and provide new empirical evidence and theoretical reference for promoting managers to implement the interactive control effectively.

The structure of this paper is as follows: Section 2 provides the theoretical background and carries out the literature review. Section 3 develops the hypothesis and explains the theoretical model. Section 4 introduces the research methods. Section 5 shows the data analysis and results. Section 6 discusses the results. Section 7 displays the conclusions of this study. Finally, Section 8 explains the limitations of this paper and the prospect of the research.

## 2. Literature Review

### 2.1. The Psychology Theory in Contingency Theory

The innovation effect of management control is not only related to the user’s own style [25], but also depends on the individual’s perception of control, that is, the intelligibility of the control practice itself [26]. Therefore, in the study of management control, the psychology theory in contingency theory is widely used [26]. Psychological contingency thought refers to those based on traditional contingency thought; particular psychological theories are introduced to take into account psychological variables, such as cognition, motivation, and their relationship with behaviors. This thought can be used to explain why there is no “universal” standard for management control that is suitable for all enterprises. Previously, some scholars have drawn on psychological theory to study the effects of budget control [27,28]. In particular, Hopwood [29] examined the impact of different use styles of accounting information on the performance evaluation of managers in cost centers through role theory. In addition, a large number of studies also draw on role theory to study the role of role stress (role ambiguity and role conflict) in the relationship between control practices and various outcomes [26]. Thus, role stress is often used as an important psychological variable to explain and predict the effect of management control [26]. According to the job demand-resource (JD-R) model, as an important aspect of stress, role stress has been widely used in previous studies to study the effect of budgets and other management and control methods [9,26]. At the same time, individuals’ job resources also determine their judgment, decision-making, and behavior [15]. Therefore, the purpose of applying psychological theory to the study of the interactive budget is to explain the operation and effect of the interactive budget at the individual level by studying the relationship between interactive budgets and individual psychological states and behaviors [30,31]. Contingency psychology effectively criticized the homogeneity hypothesis of the enterprise and further interpreted the positive influence of psychological capital on the formation of sustainable competitive advantage of the enterprise.

### 2.2. Job Demands-Resources (JD-R) Model

The job demand-resource (JD-R) model is often used to show the interaction between job demands and resources, and their impact on job performance [32], being the mainstream theoretical framework for studying the impact of work characteristics on occupational mental health [33]. According to the JD-R model, job demands mean that individuals exert physical and psychological efforts related to physical and mental consumption [34], such as role ambiguity and role conflict [35,36]. Job demands can consume individual resources and affect work performance [15]. Job resources, such as information and autonomy, can reduce the consumption related to job demands, stimulate personal learning and development, and achieve their goals [37]. In addition, according to Van den Broeck et al. [38], job demands can be divided into hindering demands and challenging demands. This difference leads to different behavioral consequences for individuals by influencing their assessment of job demands [39] and their own psychological requirements. Hindering demands are regarded as constraints or obstacles for individuals to achieve their goals, which will consume energy and interfere with achieving individual and enterprise goals. These demands are generally manifested as role stress [35,36], job insecurity [39], and others; whereas, regarding challenging demands, although they consume individuals’ energy, they can also stimulate individuals’ curiosity and motivation, motivate individuals to cope with challenges and achieve goals, generally being presented as workload [40], time stress [41], and others.

A study by Bedford et al. [9] viewed the job demands that managers may face under the traditional budget environment as hindering demands. In budget work, the enterprise’s goals or superiors’ expectations may aggravate the conflicts between managers’ accomplishing budget tasks and achieving innovation [42], bring managers additional job stress [43], and consuming managers’ resources. In addition, when the enterprise’s budget is adjusted, decisions under the traditional budget will be more centralized [44], and managers may feel uncertain about their superiors’ expectations or not know how to implement the budget due to limited information or other factors. If managers’ job resources are inadequate, they may reduce their work engagement, and innovation performance will be affected [10,15].

### 2.3. Interactive Control in Budget Management and Its Effects

Interactive budgets originate from the concept of “interactive control” proposed by Simons [13]. Simons [13] believed that there would be differences in the way managers use control systems, such as budgets. When they regularly participate in subordinates’ decision-making and carry out “face-to-face” communication on key issues to find new opportunities to deal with strategic uncertainties, this use was called interactive control [13]. The traditional budget model, with its emphasis on control of deviations and the pursuit of short-term financial goals, has been problematic in its application [14]. Libby and Lindsay [14] believed that the emphasis on results assessment in the traditional budget not only weakened the adaptability of the budget to the dynamic environment, but also led to the “budget game” behavior of subordinates to achieve the budget goal. Moreover, centralized decision-making under the traditional budget limits individuals’ access to information [8], resulting in stress that consumes their resources and weakens their motivation [15]. However, the interactive budget is problem-oriented. It can help managers obtain sufficient information through regular participation in subordinate decision-making and “face-to-face” communication, which are conducive to coping with the uncertainty of budget work [13]. In this type of budget control, managers can focus on the potential risks to the budget, with extensive and thorough discussion and learning within the organization in response to uncertainties, identifying options, and developing consensus, so that organizational rules and assumptions can be adjusted to accommodate changes in the external environment [13]. Therefore, some scholars believe that it is necessary to study interactive budgeting and its effects [44,45].

Under interactive budgets, the superior did not take the budget implementation as the only criterion for evaluating the members’ performance, reduced the “budget game” behavior [14], and promoted the continuous improvement of enterprise performance; at the same time, this budget model enables managers to focus on priorities such as uncertain factors, which is conducive to the formation of emergency strategy [46]. The environment of constant discussion and encouraged doubt under the interactive budget, on the one hand, keeps managers focused on the strategic objectives of the enterprise, which is conducive to improving organizational performance [47]; on the other hand, it stimulates managers’ creativity and initiative, helps the enterprise form higher quality decisions, and improves enterprise effectiveness [19]. Also, the interactive budget can help managers solve short-term goal stress. For example, Osma et al. [48] found that managers can identify, evaluate, and select the behavior of real earnings management by interactively using the management control system to alleviate short-term financial stress. In addition, the interactive budget can help enterprises better cope with the changing business environment. Henri [16] believed that under the interactive budget model, regular and frequent dialogue and communication between superiors and subordinates improved enterprises’ learning and innovation ability. When an enterprise uses a control system interactively, it can not only promote product innovation [23], but also increase the positive impact of product innovation on enterprise performance [44]. Furthermore, under the situation of technological turbulence, interactive control shows a more positive role in promoting enterprise innovation [3].

### 2.4. Role Stress and Its Influencing Factors

Role stress is a negative perception caused by individuals’ inability to understand or meet their role-related expectations. It is an important component of job stress, including role ambiguity and role conflict dimensions [49]. Some studies also include role overload into the dimension of role stress [50], but there are few studies specifically examining role overload [51]. According to some studies, role overload is actually a kind of role conflict [32]. Therefore, this study suggests that role stress includes two dimensions: role ambiguity and role conflict, to be consistent with previous studies [9,32]. Role ambiguity refers to an uncertain feeling caused by the lack of information related to one’s behavior expectation; role conflict refers to individuals who cannot meet multiple goals from their superiors or work tasks conflict with their values.

At present, the research on the factors influencing role stress mainly involves two levels: individual and organization. On the personal level, Kim et el. [52] found that the individual’s tolerance for role stress was affected by gender and their positions; that is, the role stress of female employees and supervisors had a significantly stronger impact on job satisfaction than that of male employees and non-supervisors. Role stress is also related to the length of service [53] and age [54]. Moreover, individuals’ negative emotions will aggravate their role conflict [55], while factors such as mindfulness level [56], resilience strength [57], and perceived importance of events [58] have a certain mitigating effect on their role stress. On the organizational level, Coelho et al. [59] found that individuals’ role stress would be affected by their relationships with their superiors. Factors such as a highly formalized management and control model [60] and tightened budget control [9] will increase the role stress of enterprise members. Enterprises can reduce the members’ perception of role stress by increasing their professional identifications [61], or by adopting more flexible budget control methods, such as enabling budget [12]. Also, factors such as the moral atmosphere of the enterprise [62] and an approachable leadership style [63] will also reduce the level of personal role stress.

## 3. Hypotheses Development and Theoretical Model

### 3.1. Hypotheses Development

#### 3.1.1. Interactive Budget and Enterprises’ Innovation Performance

The interactive budget can provide managers with the information resources they need to improve the enterprises’ innovation performance. According to JD-R theory, the organization’s demand for innovation will affect the individual’s work motivation and create job stress [15]. The interactive budget provides an open and face-to-face communication platform for enterprise members to obtain a large amount of accurate and transparent budget information. On the one hand, when preparing budgets, managers can obtain business information from the front line of the enterprise through the interactive budget, and prepare budgets based on business, so that enterprise resources can better match the innovation demands of the enterprise; managers can more clearly and timely understand the potential risks in the implementation of enterprise budget management, as well as the temporal resource status and demands of each business department, to coordinate the budget management behavior as a whole; managers can also break through the limitation of basic budget data to better link the budget with the enterprise strategy, and to better serve the enterprise innovation. On the other hand, when implementing the budget, managers can obtain information through an interactive budget, without sticking to the set budget goals, to flexibly adjust the management mode and product strategy according to environmental changes, thus improving the innovation performance of enterprises. In addition, the atmosphere of cooperation and information sharing under the interactive budget is conducive to the processing and absorption of information by managers, so that the information obtained can be truly and accurately reflected in the budget, and budget information can in turn be better understood and used by managers. In this way, the utilization rate of information resources is improved, and the innovation performance of enterprises is promoted [64].

The interactive budget can meet managers’ cognitive demands and improve enterprises’ innovation performance. According to Cacioppo and Petty [65], individuals’ cognitive demands are expressed in their satisfaction with participation and learning. Firstly, regular and adequate communication, and learning within the enterprise regarding budget uncertainties, promotes knowledge sharing and joint growth among enterprise members, which enhances the creativity of managers [66] and maximizes the overall innovation potential of the enterprise. Secondly, in the process of communication with employees, managers can timely and comprehensively understand the risks and opportunities faced by the enterprise budget management, challenge existing budget data and information, increase budget participation and advice, and have a positive impact on organizational innovation performance [67]. In addition, the interactive budget enables managers to focus on the uncertain factors that threaten budget management without consuming too much of their cognitive resources. Through constant communication on budget risk and other key issues, managers’ knowledge and skills can be enhanced and learning enthusiasm can be stimulated, encouraging managers to actively seek innovation opportunities to solve problems, which is conducive to promoting enterprises’ innovation.

Attribution theory holds that individuals’ initial view of things will affect their cognition and behavior [35]. The interactive budget provides managers with the information and cognitive resources they need and enhances their ability to budget. When faced with unexpected events, such as budget adjustments, managers under the interactive budget model are more likely to view the budget task as a challenge rather than a hindrance. This would stimulate managers’ initiative and creativity to try to meet the innovation demands of the enterprise [37]. Based on the above analysis, this study proposes the following hypothesis.

**Hypothesis** **1.***The interactive budget has a positive impact on enterprises’ innovation performance*.

#### 3.1.2. Interactive Budget and Managers’ Role Stress

The interactive budget can alleviate the role conflict of managers, thereby reducing their role stress. First of all, under the interactive budget, managers can obtain sufficient and effective business information resources by highly participating in the budgeting process of each department [16,68], and the “face-to-face” communication mode can better understand the resource demands of each business department, which is conducive to more recognition of the budget objectives reported by each department, thus reducing the conflicts caused by the inconsistency of superior and subordinate objectives. Secondly, under the interactive budget, enterprises regularly discuss and learn about key budget issues (such as the reasons why budget goals are not achieved, etc.). This not only promotes rapport between managers and subordinates, but also improves managers’ ability to meet multiple budget goals, thus alleviating role conflicts. Finally, the increased focus on a comprehensive evaluation of members’ performance under interactive budgeting alleviates the “gaming” of resources between managers and subordinates over budget targets [69]. Then, both managers and subordinates can carry out budget works according to the actual business situation of the enterprise, which promotes the consistency of members’ goals and alleviates individuals’ role conflicts.

The interactive budget can reduce role ambiguity for managers, thereby reducing their role stress. Firstly, managers can obtain frequent feedback and transparent business information through “face-to-face” communication mode [16,68], which, combined with their focus on risk factors, are conducive to managers clarifying budget targets based on the resource situation of each business department, thus reducing role ambiguity. Secondly, by using the interactive budget, managers pay more attention to the potential risk factors of the budget, which save their cognitive resources. Then they can make plans and allocate resources according to budget goals and business conditions to achieve better budget goals. Finally, interactive budgeting allows managers to increase their involvement in subordinate decision-making, as well as to discuss and learn about the reasons for deviations from budget targets, so that they can reasonably predict the outcome of budget execution and proactively adjust resource allocations. This enables greater value to be derived from the budget, which also alleviates the role blurring of managers.

To sum up, interactive budget not only helps managers identify budget goals and enhances their ability to accomplish goals, but also helps them clarify their goals, specify the way to achieve goals, and estimate the consequences of achieving goals, thus reducing managers’ role stress. Based on the above analysis, this study proposes the following hypothesis.

**Hypothesis** **2.***The interactive budget has a negative impact on the role stress level of managers*.

#### 3.1.3. The Mediating Role of Manager’s Role Stress

Previous studies have shown that role stress, as a hindering stressor [40], will negatively affect enterprise innovation performance [10,70,71]. Role stress reflects the innovation demands of the enterprises [15], which requires managers to respond with adequate personal resources. However, under the traditional budget, it is difficult for managers to obtain sufficient information and cognitive resources [72,73], which is not conducive to their investment in enterprise innovation activities. Meanwhile, traditional budget emphasis on budget goals will aggravate the negative impact of role stress on managers and pose great challenges to improving enterprise innovation performance. Role conflict can lead to managers’ self-doubt and affect their innovation awareness and execution; role ambiguity can confuse managers, leading to their inability to devote enough time and energy to innovative work [74].

From the above analysis of the relationship between interactive budget, managers’ role stress, and enterprise innovation performance, it can be seen that interactive budget probably further promotes enterprise innovation performance by influencing the level of managers’ role stress. In the interactive budget, various departments can fully communicate and learn, cooperate, and share information with each other; managers can improve their problem-solving ability and obtain the information they need through continuous vertical and horizontal communication, which can alleviate their role conflict and role ambiguity, thus reducing the managers’ sense of demand for work resources and allow them to have more sufficient resources for enterprise innovation activities. Meanwhile, managers’ information and cognitive resources can be satisfied under the interactive budget; then, they will more actively participate in the innovation work [63,64]. Based on the above analysis, this study proposes the following hypothesis.

**Hypothesis** **3.***The managers’ role stress plays mediating role in the relationship between interactive budget and enterprise innovation performance*.

#### 3.1.4. The Moderating Role of Environment Turbulence

Environmental turbulence refers to the frequency of unpredictable and highly diversified events in the enterprise environment [75], with strong uncertainty. According to previous studies, this uncertainty is generally related to customer preference, technological shift, and the degree of competition [76,77,78]. Contingency theory suggests that the innovation effect of budget management will be interfered with by environmental factors [79]. That is, compared with a stable environment, a more turbulent environment will increase managers’ resource demands [80], and enterprises’ emphasis on the improvement of dynamic [81] and innovative capabilities [82].

The interactive budget can match the demands of managers and enterprises in a turbulent environment. On the one hand, under the interactive budget, managers keep communicating and widely participate in budgeting, making budget information more transparent and managers’ resource demands able to be met, reducing their perception of uncertain environments [83]. This means that managers can clarify their budget responsibilities, take better initiative, and promote enterprise innovation performance. On the other hand, the interactive budget makes managers focus on uncertain information related to potential budget risks, which enhances the enterprise’s ability to perceive uncertain environments. Then, this information will be actively discussed and learned by managers to determine plans and develop a consensus, transforming the personal perception of threats and opportunities into the enterprise’s ability to cope with the turbulent environment. As the dynamic capability model proposed by Pavlou and Sawy shows [84], the interactive budget will improve enterprises’ dynamic capability, thus improving enterprises’ innovation performance. Based on the previous analysis, it can be predicted that in a more turbulent environment, enterprises will use the interactive budget more actively and frequently, thus increasing the role of the interactive budget in promoting enterprise innovation performance. However, it has also been shown that when considered from an individual perception perspective, in uncertain environments such as crises, individuals may experience panic and avoidance behaviors [85], increase the need for cognitive closure [86], and are less willing to access information. When considered from the above perspective, the results may appear in the opposite direction. Based on the above analysis, this study proposes the following hypothesis.

**Hypothesis** **4.***Environmental turbulence has a moderating role in the relationship between interactive budget and enterprise innovation performance*.

### 3.2. Model Specification

The model in this study is based on contingency theory. Firstly, based on the psychology theory in contingency theory, this study investigates the mechanism of interactive budgets on enterprise innovation performance from the perspective of managers’ role stress according to the job resource-demand (JD-R) model, a psychological theory. As mentioned above, the JD-R model is widely used in the study of role stress [32] and has recently been applied in some studies to explain the effect of budget control [9]. Job resources and job demands will lead to different psychological processes, thus making individuals produce different behaviors. The perception of stress generated by job demands under the traditional budget will consume individuals’ job resources and hinder their innovative activities [9,15], while the interactive budget can provide individuals with job resources, make up for the loss of resources due to stress, and improve the enthusiasm and initiative of managers [15]. Second, based on traditional contingency theory, this study examines the impact of environmental turbulence on the effectiveness of innovation under interactive budgets. In turbulent environments, managers’ resource needs are difficult to meet under traditional budgets [15], while interactive budgets can provide them with substantial resources to adapt to changes in the environment [80]. The research model is shown in Figure 1.

## 4. Method

### 4.1. Sample and Procedure

This research conducted an online cross-sectional survey, and all questionnaires are distributed through the reliable online platform Credamo (www.credamo.com) during the data collection process. The participants are CEOs, CFOs, financial managers, and department managers with financial work experience from enterprises in the Yangtze River Delta region. They have decision-making power and rich budget work experience. As an important development strategy, the integration of the Yangtze River Delta bears an important responsibility for the implementation of the national innovation strategy in the new era, it also has made important contributions to the continuous increase of China’s economic contribution rate. In addition, the “Evaluation Report on China’s Regional Innovation Capability 2021” showed that the “three provinces and one city in the Yangtze River Delta” are all in the top 10 list of innovation capacity in China. These are the reasons why we selected samples in the Yangtze River Delta.

In order to ensure that only qualified respondents are able to fill in our questionnaire, the questionnaire is restricted to those who meet the conditions before they can formally fill in the questionnaire. To avoid the common method bias problems, a simple pretest was carried out in the formal investigation to evaluate the appropriateness of the questionnaire’s words and the rationality of the indicator set [87]. We recruited 40 participants eligible for this pretest by telephone through two students of business administration who had budget work experience. These participants were asked to fill in our questionnaire (they had cooperated with the university before, so they showed high cooperation). We sent the questionnaire links to them, and 38 responses were collected, then, we slightly revised the questionnaire according to the pretest results. In the formal survey, after excluding 40 responses from the pretest, we distributed a total of 270 questionnaires. In the questionnaire, it is stated that “the information filled in will be strictly confidential and will not be transmitted to the public”, so that the respondents can fill in with confidence; moreover, to avoid repeated answering, we limited each IP address to fill in the questionnaire only once. All questionnaires are anonymous. For the 270 questionnaires recovered from the formal survey, 228 valid data were finally retained after removing the samples with missed answers, too short/too long answer time, and too high a consistency of answers (the effective recovery rate is 84.44%). The demographic characteristics of the sample are shown in Table 1.

### 4.2. Measures

In this study, five potential variables were measured with a mature and widely used scale. To ensure the validity of the scale, we conducted translation-back translation processing on all scales. The questionnaire uses a 7-level Likert scale (1 means “strongly disagreed” and 7 means “strongly agreed”). Moreover, the data were analyzed by using IBM SPSS 23.0 (IBM, Amonk, New York, USA), IBM SPSS Amos 24.0 and Mplus 7.4 (Linda Muthén & Bengt Muthén, http://www.statmodel.com/, accessed on 8 November 2022).

#### 4.2.1. Interactive Budget

Referring to Henri [16] and Chong et al. [19], this research used seven items to evaluate the use of enterprise’s interactive budget. The scale focuses on whether managers and employees can often talk, discuss, and reach a consensus on budget problems, and form a unified form of expression, whether managers and employees can focus on and understand the common problems in budget management, and whether managers and employees can work together on budget work. Cronbach’s alpha coefficient of the scale is 0.879.

#### 4.2.2. Role Stress

The measurement of role stress (i.e., role conflict and role ambiguity) is based on the research results [49]. In previous studies, the scale has shown a good measurement effect [88,89]. Meanwhile, referring to Bedford et al. [9], we deleted one item with a low standardized factor load in the original role ambiguity scale to ensure that the scale has sufficient reliability and validity, which is “I understand the expectations of leaders in my work”. Finally, 9 items are used to measure the role stress level of managers. For role conflict, we measure it through four items: “I will be asked to complete some different work”, “I will receive contradictory task arrangements”, “I will do some redundant work”, and “My budget work will be praised and criticized by people around me”; Role ambiguity is measured by five items: “I don’t know what I can do in my work”, “I don’t know my responsibilities in my work”, “I can hardly reasonably allocate time in my work”, “I don’t have a clear work goal”, and “I don’t know what leaders expect of me in my work”. Cronbach’s alpha coefficient of the scale is 0.909.

#### 4.2.3. Innovation Performance

Innovation is a dynamic and multidimensional concept, representing a process of change [90]. Tavassoli and Karlsson [91] used four main aspects to measure innovation according to OECD [92]: product, process, organization, and marketing innovation. Therefore, the scale of our study, learned from Ritalin et al. [93], for measuring innovation performance included four items. By asking the interviewees, compared with the major competitors in the same industry, the enterprise’s innovation performance in the past three years was evaluated in terms of new products or services, production processes, management, and marketing practices. Cronbach’s alpha coefficient of the scale is 0.810.

#### 4.2.4. Environment Turbulence

Referring to the scale developed by Miller and Friesen [94], this study used five items to measure the degree of environmental turbulence. These five items evaluate the environmental turbulence from the enterprise’s marketing strategy, the update speed of products or services, the update speed of technology, the actions of competitors, and the customer’s demand preference. This scale is often used to measure the dynamics of the environment in subsequent studies [95,96]. Cronbach’s alpha coefficient of the scale is 0.836.

#### 4.2.5. Control Variables

Considering the characteristics of China’s economic transformation, the innovation effect of enterprises will be affected by their property rights nature [97]; the research of Chandy and Tellis [98] showed that the innovation capability of enterprises varies with their scale. Therefore, we chose the nature of property rights and the scale of the enterprise as the control variables.

## 5. Results

### 5.1. Common Method Bias Test

Harman’s single-factor test was conducted by using IBM SPSS 23.0. The results showed that the cumulative variance contribution rate was 67.134%, and the variance of the first factor accounted for 34.740% of the total variance (below the threshold of 40%), indicating that the probability of common method deviation in this study was low. In addition, we also used Mplus 7.4 for confirmatory factor analysis to further confirm the results of the Harman test. Table 2 shows that the fitting results of the four-factor model (χ^2^/df = 1.676, RMSEA = 0.063, SRMR = 0.057, CFI = 0.944, and TLI = 0.936) are significantly better than those of the single factor model (χ^2^/df = 4.915, RMSEA = 0.138, SRMR = 0.126, CFI = 0.671, and TLI = 0.631). These two tests together show that common method bias should not be the main problem in our research.

### 5.2. Reliability and Validity Test

As previously mentioned, Cronbach’s alpha coefficient is between 0.810–0.909, indicating that the scale has good reliability. About convergence validity, first of all, the results of confirmatory factor analysis in Table 2 show that χ^2^/df = 1.676, RMSEA = 0.063, SRMR = 0.057, CFI = 0.944, and TLI = 0.936, indicating that the model is fully fitted; furthermore, Table 3 shows that the AVE values of each latent variable are between 0.505–0.521, both of which are greater than 0.5. Both of them together show that the scale has good convergence validity. Secondly, as shown in Table 2 and Table 3, compared with other models, the fitting results of the four factors model (χ^2^/df = 1.676, RMSEA = 0.063, SRMR = 0.057, CFI = 0.944, and TLI = 0.936) are significantly better than those of other models, and the square root of AVE of each latent variable is greater than the correlation coefficient of this latent variable and other latent variables. These showed big differences between the four constructs in this study. Finally, as mentioned above, the scales used in this study are all mature scales, which are widely used; moreover, with the help of relevant experts, we made appropriate adjustments based on the research background, object, and purpose, so it has good content validity.

### 5.3. Descriptive Statistics

The average value, standard deviation, and correlation coefficient of each variable are shown in Table 3. It can be seen from the table that there is a significant negative correlation between the use of interactive budget and the level of manager role stress (r = −0.363, *p* < 0.001), and a significant positive correlation between the use of interactive budget and innovation performance (r = 0.427, *p* < 0.001); there was a significant negative correlation between manager role stress and enterprise innovation performance (r = −0.396, *p* < 0.001). The results are consistent with the above hypotheses.

### 5.4. Hypothesis Testing

Table 4 shows the non-standardized regression results of the main variables. The results show that the interactive budget has a significant positive impact on enterprise innovation performance (β = 0.529, *p* < 0.001), which has a significant negative impact on the role pressure of managers (β = −0.343, *p* < 0.001), meaning H1 and H2 were supported. The results also showed that, according to the regression results, the role stress of managers has a significant negative impact on innovation performance (β = −0.350, *p* < 0.001), which provides preliminary support for our mediation hypothesis of role stress.

### 5.5. Mediating Effects Testing

Based on the suggestions of Sobel [99] and Hayes [100], this study used the method of bias-corrected non-parametric percentile Bootstrap method to conduct 228 data through Amos 24.0, testing the mediation role of managers’ role stress in the relationship between interactive budget and enterprise innovation performance, and obtaining the indirect effect path coefficient, standard deviation, significance, and 95% confidence interval of model path analysis. The results are shown in Table 5. The results show that the |Z| of the path of interactive budget influencing innovation performance through role stress is greater than 1.96, and the 95% confidence interval of this path does not include 0, indicating that the mediating effect is significant. Therefore, H3 is supported.

### 5.6. Moderating Effects Testing

This study used the hierarchical regression method to test the moderating effect of environmental turbulence. To eliminate the influence of collinearity, this paper centralized the two variables when constructing the product term of the independent variable and moderating variable, and the results are shown in Table 6. According to the models Model2 and Model3, the interaction between the interactive budget and environmental turbulence has a significant positive impact on enterprise innovation performance (β = 0.223, *p* < 0.001). The main reason for the positive direction of H4 could be the sample characteristic. We distributed questionnaires in the Yangtze River Delta region, where “the three provinces and one city in the Yangtze River Delta” are all in the top 10 in China in terms of innovation capability level. The interviewees are CEOs, CFOs, finance managers, and department managers from companies in the Yangtze River Delta region who have experience in finance, have decision-making power, and have more experience in budgeting. Therefore, as mature managers, during times of environmental turbulence, they may be less likely to retreat from interactive budget efforts and more likely to take a proactive approach. To show the moderating effect of environmental turbulence more intuitively, we conducted a simple slope analysis, as shown in Figure 2. It can be seen from the figure that when the degree of environmental turbulence is high, the use of the interactive budget model has a more significant positive impact on enterprise innovation performance; when the degree of environmental turbulence is low, the relationship between the interactive budget and enterprise innovation performance is not significant. This was in line with our expectations. Therefore, H4 is supported.

### 5.7. Further Analysis

The innovation decisions of enterprises and the role stress level of managers will be affected not only by personal factors such as gender [52], but also by industrial factors such as organizational knowledge and technology level [93]. Therefore, this study further examined the impact of interactive budgets on enterprise innovation performance and role stress of managers based on two sample characteristic variables, industry category (high-tech enterprises and non-high-tech enterprises) and manager gender (male and female). The results showed that there is a significant difference between male and female managers in the impact of interactive budgets on enterprise innovation performance (*p* < 0.05); there is a significant difference between high-tech enterprises and non-high-tech enterprises (*p* < 0.05) in the impact of interactive budgets on the level of managers’ role stress. Specifically, in the process of interactive budgets promoting enterprise innovation performance, female managers (β = 0.763, *p* < 0.001) contribute more than male managers (β = 0.495, *p* < 0.001); interactive budgets in high-tech enterprises (β = −0.719, *p* < 0.01) have more negative effects on the managers’ role stress than that of non-high-tech enterprises (β = −0.335, *p* < 0.001). The results are shown in Table 7.

## 6. Discussion

Some scholars emphasized that it is necessary to further study the impact of management control on the relationship between innovation and performance [17,88], and this study responded to their call by examining the relationship between interactive budgets and enterprise innovation performance, as well as exploring the mechanism between this relationship from the perspective of managers’ role stress, and studying the impact of environmental turbulence on this relationship.

### 6.1. Theoretical Implications

First, this study expands the literature on influencing factors of innovation performance. (a) Although previous studies have shown the effect of interactive budgets on enterprise innovation [24,46], little research explored the mechanism between them. This study examined the effect mechanism of the interactive budget on enterprise innovation performance from the psychological level of managers. We elaborated on the effect path of the interactive budget on enterprise innovation performance based on the JD-R model, according to the psychology theory in contingency theory, which further opens the “black box” of the interactive budget and enterprise innovation performance. (b) This study also finds that the innovation effect of the interactive budget is more significant in enterprises with more female managers. Compared with men, women are more sensitive to the opportunistic behaviors that could bring risks to budget work [101] and are not prone to overconfidence [102], which will prompt women to identify risks more quickly and cooperate effectively with other members, thus coping with the uncertainty of budget work. This finding highlights the positive role of gender factors in the research of enterprise innovation performance. (c) The results show that the promoting effect of the interactive budget on enterprise innovation performance is more significant in a relatively turbulent environment. This result verifies and expands the previous findings [2,3] that the demand for resources in a turbulent environment makes interactive budgets more effective, which complements the positive implications of interactive budgets in coping with external contexts.

Second, this study enriches the relevant research on role stress. In the field of management control, previous studies on role stress mostly focused on the traditional budget [9], this study adds the antecedent variables affecting the role stress under the interactive budget. Our results indicated the negative correlation between interactive budgets and the stress of managers’ roles, which verifies the previous studies [10,71], that is, the managers’ role stress is not conducive to enterprise innovation. Additionally, our result showed that in high-tech enterprises, the interactive budget has a stronger inhibitory effect on the managers’ role stress, which highlights the important influence of industry factors on individual role stress. High-tech enterprises are facing a more competitive market environment, leading enterprises to increase access to information and frequency of information searches [80], resulting in the active use of interactive budgets. Managers in high-tech enterprises generally have good educational backgrounds, with strong knowledge absorption and information processing abilities, and are able to understand and use the job resources provided by interactive budgets more efficiently and quickly, thus strengthening the alleviating role of interactive budgets on managers’ role stress.

Third, this study supplies literature on the impact of interactive budgets on individual psychology. Previous studies on the interactive budget are mostly related to individual use styles [19,22,103], with little consideration of individual perceptions of budget management. The results of this study showed that the interactive budget has a significant inhibitory effect on the level of managers’ role stress, which means that the interactive budget can cope with the resource consumption of the individual due to the job demand [15]. This result provides a theoretical basis for solving the psychological problems faced by managers under traditional budgeting, and this is consistent with previous studies [9].

### 6.2. Practical Implications

First, enterprises need to emphasize the role of the interactive budget when carrying out innovation activities. In the competitive environment with rapid technology iteration and fierce market change, managers and employees should actively interact and share information on factors such as their strategic understanding, key business activities, resource requirements, and departmental value propositions in all aspects of budget management. Thus, it can stimulate managers’ creative thinking and meet managers’ resource needs for innovative work. Also, budget activities should be business-oriented and the preparation and execution of budgets should be flexibly adjusted through interaction. Thus, managers can focus more on the risk factors regarding budget target deviation through the interactive budget, make the budget better serve enterprises’ resource allocation, and promote the improvement of enterprises’ innovation ability.

Second, enterprises need to pay attention to the psychological factors of managers when conducting budget management. To begin with, in order to keep managers actively working in the budget, enterprises need to strengthen cross-departmental communication to open information channels. Through feedback from the activities of the budget departments, enterprises can understand their resource needs and make adjustments. Then, enterprises should guide managers to deeply understand and agree with the organizational strategy and business objectives. Enterprises can make managers clear the responsibilities related to their role expectations by strengthening business skills training, reducing the managers’ role stress in budget activities, and improving their budget working ability. Finally, the enterprise should establish a transparent and reasonable performance indicator system matching job responsibilities to assess managers’ budget activities effectively. This assessment content should emphasize not only cash rewards, but also spiritual rewards, to ensure the transparency and fairness of the assessment process.

Third, female managers should value the use of interactive budgets. For enterprises, they can consider promoting the use of interactive budgets by women managers, that is, having more female managers regularly communicate the company’s strategy and business objectives to all business departments, which can promote the formation of strategic synergies within the enterprise and individual innovation initiatives. For female managers, they can regularly initiate “face to face” communication activities on key budget issues, such as budget standards and procedures, determination of budget target values, and reasons for not achieving budget goals in order to collect information from various business departments to set budget goals in line with the actual situation of the enterprise. Also, in the process of budget implementation, female managers should pay more attention to the tracking individual behaviors and acquire internal and external environment information of the enterprise, to identify risks and opportunities, thus adjusting the budget to ensure the realization of the enterprise strategy. In addition, enterprises should actively promote the cooperation between male and female managers, take advantages of female managers in the interactive budget, and try to avoid the traditional male-dominated phenomenon in strategic choice, innovation decision-making, and other aspects.

Fourth, high-tech enterprises should realize the role of the interactive budget in guaranteeing managers’ psychological health. Since the COVID-19 pandemic, enterprises have been operating in a more turbulent environment, and the high-tech industry has a greater demand to innovate than others, with greater stress (i.e., technological stress, etc.). Managers should regularly explain the strategic orientation of the enterprise to subordinates, identify benchmark enterprises in the industry, and keep abreast of the research and development, capital, and other demands of various departments. These are not only conducive to promoting the consistency of goals between managers and employees, but also enable managers to clarify budget objectives, guide budgeting based on the actual business situation of the entire enterprise, and relieve their role stress. In addition, managers are supposed to attend regular management training, focusing on the latest policy information in the technology industry, so that they can make targeted plans, enhance their ability to meet budget targets, and relieve their psychological stress caused by the uncertain environment.

## 7. Conclusions

This study examined the relationship between the interactive budget and enterprise innovation performance in the Yangtze River Delta region of China, exploring the mechanism between this relationship from the perspective of managers’ role stress and investigating the impact of environmental turbulence. Our results show that: (1) the interactive budget has a significant positive relationship with enterprise innovation performance. The interactive budget provides managers with information and cognitive resources, and managers will be more actively involved in innovation activities, thus improving enterprise innovation performance. (2) The impact of the interactive budget on enterprise innovation performance is partially mediated by managers’ role stress. Managers can relieve the role stress through the job resources provided by the interactive budget so that they can respond to the innovation demand of enterprises with more adequate resources and promote the innovation performance of enterprises. (3) Environmental turbulence positively moderates the relationship between the interactive budget and enterprise innovation performance. Managers in turbulent environments have increased demands for resources, and the interactive budget can provide resources for them. Therefore, enterprises will use interactive budgets more actively in a turbulent environment, thus further improving innovation performance. (4) The positive relationship between interactive budgets and enterprise innovation performance is more significant in enterprises with more female managers. Being sensitive to uncertain information and good at cooperation, female managers can promote the greater promoting effect of interactive budgets on enterprise innovation performance. (5) The negative relationship with the role stress of managers is more significant in high-tech enterprises than in other types of enterprises. Managers in high-tech enterprises often have good educational backgrounds, which will improve the utilization rate of resources provided by interactive budgets, thus relieving their role stress more effectively.

## 8. Limitations and Future Research

This study has the following limitations. First, we collected cross-sectional data. Although Harman single-factor test showed that common method bias will not affect our study, it is difficult to determine the causal relationship between variables. Therefore, future research can consider the causal relationship between variables based on longitudinal data. Second, there are many factors that affect the main variables in this study, including some individual-level variables, such as the manager’s age and length of service, but this study only controls for enterprise-level variables. Therefore, future research can include other potentially influencing factors in the research framework. In addition, this study only considers the effect of gender and industry factors on the interactive budget, the managers’ role stress, and the enterprise innovation performance. To enrich this field, future studies could consider other macro factors, such as the company’s life cycle [104]. Finally, this study discusses the impact of management and control methods on innovation only from the budget level. Future research could examine the impact of management on control practices at other levels (i.e., cost control, informal control, etc.), or explore the mediation role of other psychological variables.

## Figures and Tables

**Figure 1 ijerph-20-02190-f001:**
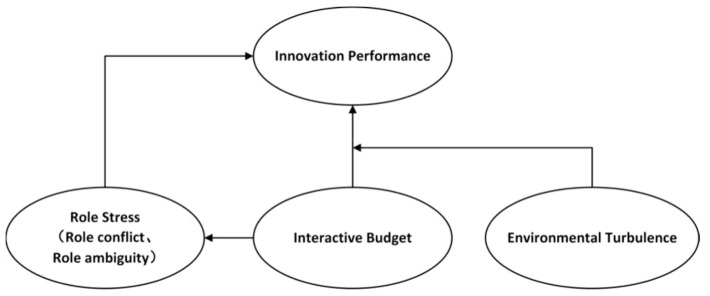
Research model.

**Figure 2 ijerph-20-02190-f002:**
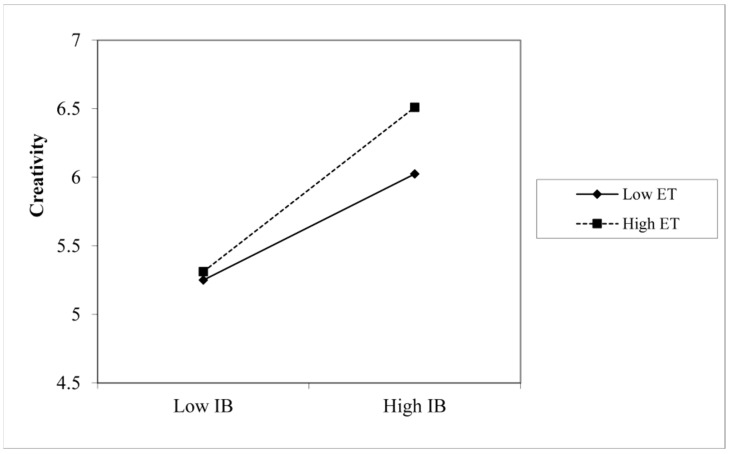
Moderation of environment turbulence on the relationship between interactive budget and innovation performance.

**Table 1 ijerph-20-02190-t001:** Demographic characteristics of the sample.

Characteristic	Number	Percent
Industry type		
High-tech	47	20.6
Non-high-tech	181	79.4
Property right nature		
State enterprise	165	72.4
Non-state enterprise	63	27.6
Number of employees		
Under 20	7	3.1
21 to 300	74	32.5
301 to 1000	101	44.3
Over 1000	46	20.2
Gender		
Male	79	34.6
Female	149	65.4
Age (years)		
Under 25	40	17.5
26 to 36	149	65.4
Over 36	39	17.1

**Table 2 ijerph-20-02190-t002:** Confirmatory factor analysis.

	χ²/df	RMSEA	SRMR	CFI	TLI
Four-factor model IB, RS, IP, ET	1.676	0.063	0.057	0.944	0.936
Three-factor model IB + RS, IP, ET	2.385	0.086	0.075	0.885	0.870
Two-factor model IB + RS + IP, ET	4.302	0.128	0.117	0.723	0.689
Single-factor model IB + RS + IP + ET	4.915	0.138	0.126	0.671	0.631

Notes: IB, interactive budget; RS, role Stress; IP, innovation performance; ET, environment turbulence; +, merge into a factor.

**Table 3 ijerph-20-02190-t003:** Mean, SD, correlation coefficients, and the square root of average variance extracted values.

	Mean	SD	PRN	NE	AVE	IB	RS	IP	ET
PRN	1.280	0.448							
NE	2.180	0.786	0.042						
IB	5.680	0.767	0.035	0.011	0.510	**0.714**			
RS	3.380	1.202	−0.001	0.022	0.505	−0.363 ***	**0.711**		
IP	5.870	0.703	0.057	−0.027	0.521	0.541 ***	−0.396 ***	**0.722**	
ET	5.350	0.859	0.137 *	−0.099	0.515	0.631 ***	−0.238 ***	0.427 ***	**0.718**

Notes: * *p* < 0.05, *** *p* < 0.001. SD, standard deviation; PRN, property right nature; NE, number of employees; AVE, average variance extracted. Bold values on the diagonal are the square root of average variance extracted (AVE) values.

**Table 4 ijerph-20-02190-t004:** Non-standardized regression results.

	Estimate	SE	CR	P
IB->IP	0.529	0.083	6.344	0.000 ***
IB->RS	−0.343	0.077	−4.453	0.000 ***
RS->IP	−0.350	0.099	−3.536	0.000 ***

Notes: *** *p* < 0.001. SE, standard error; CR, critical ratio, the ratio of estimate to SE.

**Table 5 ijerph-20-02190-t005:** Results of the mediating effect.

	Estimate	SE	Z	Percentile		Bias-Corrected	
				Lower	Upper	Lower	Upper
IB->IP							
Total	0.648	0.111	5.838	0.475	0.907	0.469	0.894
Direct	0.529	0.113	4.681	0.338	0.783	0.333	0.775
IB->RS->IP							
Indirect	0.120	0.057	2.105	0.043	0.254	0.044	0.256

Notes: Z, the ratio of estimate to SE.

**Table 6 ijerph-20-02190-t006:** Results of the moderating effect.

	IP		
	Model1	Model2	Model3
PRN	0.059	0.040	0.024
NE	−0.030	−0.035	0.000
IB		0.540 ***	0.538 ***
IB × ET			0.223 ***
R^2^	0.004	0.296	0.344
△R^2^	0.004	0.291 ***	0.048 ***
F	0.473	31.326 ***	23.275 ***

Notes: *** *p* < 0.001. R^2^, coefficient of determination; F, variance test.

**Table 7 ijerph-20-02190-t007:** Heterogeneity test results.

	Grouping and Regression Results								Group Significance Test		
	Unstd	SE	P	Std	Unstd	S.E.	*p*	Std	DF	CMIN	*p*
	Male				Female						
IB->IP	0.454	0.120	0.000 ***	0.495	0.798	0.116	0.000 ***	0.763	1.000	3.921	0.048 *
IB->RS	−0.306	0.120	0.011 *	−0.346	−0.350	0.098	0.000 ***	−0.430	1.000	0.079	0.779
	High-tech				Non-high-tech						
IB->IP	0.478	0.171	0.005 **	0.736	0.668	0.096	0.000 ***	0.623	1.000	0.709	0.400
IB->RS	−0.720	0.231	0.002 **	−0.719	−0.272	0.077	0.000 ***	−0.355	1.000	4.592	0.032 *

Notes: * *p* < 0.05, ** *p* < 0.01, *** *p* < 0.001. Unstd, unstandardized coefficients; Std, standardization coefficient; DF, degree of freedom; CMIN, chi-square.

## Data Availability

The datasets used in this research are available upon request from the corresponding author.

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
