# Peer review of "The Impact of Interactive Control in Budget Management on Innovation Performance of Enterprises: From the Perspective of Manager Role Stress"

_ijerph, 2023, doi:10.3390/ijerph20032190_

Round 1

Reviewer 1 Report

It is my pleasure to read this fantastic paper entitled “The Impact of Interactive Control in Budget Management on Innovation Performance of Enterprises: From the Perspective of Manager Role Stress”. Authors investigated the influence of the interactive control on enterprise innovation performance from the perspective of managerial psychology, examining the mediating effect of managers' role stress and the moderating effect of environmental turbulence in this influence. The study used a questionnaire survey to verify these effects through the structural equation modelling approach, with a sample of enterprises implementing budget management in the Yangtze River Delta region of China. The following comments may help the authors to revise the paper:

1)    The abstract is perfectly explained. It will be great if authors add the basic methodological portion add here, i.e. name of population, sample size, sampling technique, SEM version name and so on. The theoretical implications are specifically mentioned. Authors may add practical implications/contributions of this paper at the end of abstract.

2)    Authors started the “1. Introduction” with the lines Innovation can enhance the flexibility and adaptability of enterprises and is a key 31 element for enterprises to acquire and maintain core competitiveness [1]. In the process 32 of enterprise innovation, management control plays an indispensable role [2].” The introduction should be started with attractive lines that explain the current scenarios/trends of the research topic. More relevant recent papers should be cited. One extra paragraph can be added on the current scenario of “Innovation Performance of Enterprises” in China. Authors should add the objective and paper organization at the end of the introduction section.

3)    More recent relevant papers should be cited here. Figure 1. Research model is easy to understand. 

4)    Authors should mention the sampling technique. Data collection process should be explained more for clear understating for readers. How surveys were distributed and who helped to manage the data collection process?

5)    The paper also needs an overarching theoretical framework to guide its hypothesis development, especially H4 which talking about moderating role Environmental turbulence in the relationship between interactive budget and enterprise innovation performance. 

6)    “6.2. Limitations of the Study” should be after conclusion. Conclusion should be explained more as per objectives and study findings.

7)    What are the contributions of your study to the current literature of innovation performance? “Implications of the study” should be added as an extra paragraph that reflects more clear idea of the contribution of the study.

8)    Authors should check and double check the in-text citations and references list.

Good-Luck

Reviewer 2 Report

The manuscript investigates the positive impact of the budgeting approach as an organizational element from an individual perspective, which is very innovative. The statistical methodology is scientific and the results of the study verified the positive side of interactive budgeting and also found specific positive aspects of interactive budgeting for gender and the specific type of enterprises, making the results enlightening. However, the following questions remain, which I hope the authors will answer.

1. In Introduction, the author should introduce what is traditional budgeting and what is interactive budgeting, especially for readers who are not familiar with the concepts of both, and authors should highlight the concepts or features and differences between them. Accordingly, in the Literature Review, I expect to see the differences between interactive and traditional budgets in various aspects. However, the authors emphasized more on the features and results of interactive budgeting rather than their differences, which prevented readers from quickly understanding the advantages of interactive budgeting.

2. In section 2.3, I feel that the theoretical logic of how different budget control methods affect individual stress is not clear enough, and the logic of the two needs to be strengthened, which is very important for the construction of the model.

3. Regarding the control variables in this study, it seems that all the variables chosen in the study are at the organizational level, why not at the individual level?

4. In Discussion, the practical implications are not clear enough. First, the manuscript does not provide sufficiently specific recommendations for specific enterprises. More industry- and firm-level information should be added to the Discussion to bridge the gap between theoretical research and practice. In addition, the discussion regarding female findings is not deep enough, for example, it is mentioned that leaders should understand the resource needs of female managers, but the results of this study do not lead to this implication.

5. There are also certain grammatical errors in the article, please check them carefully, for example: Page 3 (4) Do the interactive budget innovation and its effect on managers role stress differ by firm enterprises' heterogeneity (e.g., managerial gender, industry type)?

Reviewer 3 Report

Thank you very much for providing this opportunity reviewing this paper. I really like the topic of this paper. However, there might be some concerns need to be addressed.

1. Significance of topic. Why this topic is theoretically important? First, I am really curious about why you focus on the interactive control in budget management? Second, what is the interactive control?  Please provide more detailed explanation about this concept.

2. Theoretical perspective. Managerial psychology seems too broad to guide this manuscript. Could you please offer some specific framework to guide this model?

3.  Why you use the role stress as the moderator? Is it relevant to the budget management?

4. why environment turbulence? Please using one framework to guide you model. Now it seems too data-driven. Why environment turbulence moderates the direct effect rather than indirect effect?

5. Method. Some correlation seems too high. Cross-sectional design may bring serious common method biases. 

6. In the discussion section. Please provide more discussion with current literature.

Reviewer 4 Report

Dear authors, very nice presentation of the subject. though I have some suggestions to make.

1. you mentioned the difference existing between hi-tech or not companies. i would like to see also how the different stage of organizational life cycle for each company is related to the interactive control.

2. I have a suggestion about the model you present at line 315. in my oppinion the model should have opposite direction, since the turbulant environment is the exogenous force affecting the operation of the company and the performance of managers.

Round 2

Reviewer 3 Report

I believe that most of my comments are well addressed.